# Social support and ideal cardiovascular health in urban Jamaica: A cross-sectional study

**Alphanso L. Blake**[1,2], **Nadia R. Bennett**[1], **Joette A. McKenzie**[1], **Marshall K. Tulloch-Reid**[1], **Ishtar Govia**[1], **Shelly R. McFarlane**[1], **Renee Walters**[1], **Damian K. Francis**[3], **Rainford J. Wilks**[1], **David R. Williams**[4], **Novie O. Younger-Coleman**[1‡], **Trevor S. Ferguson**[1‡*]

**1** Caribbean Institute for Health Research, The University of the West Indies, Mona, Kingston, Jamaica, **2** School of Clinical Medicine and Research, The Faculty of Medical Sciences, The University of the West Indies, Nassau, The Bahamas, **3** School of Health and Human Performance, Georgia College and State University, Milledgeville, Georgia, United States of America, **4** Department of Social and Behavioral Sciences, Harvard T. H. Chan School of Public Health, Boston, Massachusetts, United States of America

‡ NOYC and TSF are joint senior authors on this work.
* trevor.ferguson02@uwimona.edu.jm

**Data Availability Statement:** The data used in this manuscript and associated data dictionary are available in supplementary files S1 Dataset, S2 Dataset, and S1 Dictionary.

## Abstract

Recent studies have suggested that high levels of social support can encourage better health behaviours and result in improved cardiovascular health. In this study we evaluated the association between social support and ideal cardiovascular health among urban Jamaicans. We conducted a cross-sectional study among urban residents in Jamaica's southeast health region. Socio-demographic data and information on cigarette smoking, physical activity, dietary practices, blood pressure, body size, cholesterol, and glucose, were collected by trained personnel. The outcome variable, ideal cardiovascular health, was defined as having optimal levels of ≥5 of these characteristics (ICH-5) according to the American Heart Association definitions. Social support exposure variables included number of friends (network size), number of friends willing to provide loans (instrumental support) and number of friends providing advice (informational support). Principal component analysis was used to create a social support score using these three variables. Survey-weighted logistic regression models were used to evaluate the association between ICH-5 and social support score. Analyses included 841 participants (279 males, 562 females) with mean age of 47.6 ± 18.42 years. ICH-5 prevalence was 26.6% (95%CI 22.3, 31.0) with no significant sex difference (male 27.5%, female 25.7%). In sex-specific, multivariable logistic regression models, social support score, was inversely associated with ICH-5 among males (OR 0.67 [95% CI 0.51, 0.89], p = 0.006) but directly associated among females (OR 1.26 [95%CI 1.04, 1.53], p = 0.020) after adjusting for age and community SES. Living in poorer communities was also significantly associated with higher odds of ICH-5 among males, while living communities with high property value was associated with higher odds of ICH among females. In this study, higher level of social support was associated with better cardiovascular health among women, but poorer cardiovascular health among men in urban Jamaica. Further research should explore these associations and identify appropriate interventions to promote cardiovascular health.

**Funding:** Funding for this study was provided by the Bernard Lown Scholar program, (grant number: BLSCHP-1604). Sponsor's Website: https://www.hsph.harvard.edu/lownscholars/. The grant was awarded to TSF. The funders had no role in study design, data collection and analysis, decision to publish, or preparation of the manuscript.

**Competing interests:** The authors have declared that no competing interests exist.

## Introduction

Cardiovascular disease (CVD), manifesting clinically as coronary heart disease, cerebrovascular disease, and peripheral arterial disease, remains the leading cause of morbidity and mortality globally [1, 2]. The prevalence of CVDs increased two-fold from 1990 to 2019, from 271 million to 523 million cases globally [3]. Additionally, there was a 50% increase in deaths from 12.1 to 18.6 million over the same period [4]. Social determinants of health, such as socioeconomic position, poverty, social networks, and access to health care services [5], continue to be among the major contributors to cardiovascular diseases [3]. Some experts project a further increase in the prevalence of CVD in several world regions, including the Caribbean, where there is an expected doubling in the geriatric population over the next thirty years [6].

Several risk factors for CVDs have been recognised, including seven factors identified in the American Heart Association (AHA) "Life's Simple 7" programme: high blood pressure, high cholesterol, high blood sugar, physical inactivity, unhealthy diet, overweight/obesity and cigarette smoking [7]. The AHA also reported that maintaining the recommended levels of these characteristics will reduce morbidity and mortality from CVD and other conditions [8, 9]. Having the recommended levels of these seven characteristics have been used to define the construct of ideal cardiovascular health (ICH). This concept was introduced by the AHA in 2010 to refer to the absence of CVD and normal levels of key risk factors, namely normal blood pressure, normal glucose, normal cholesterol, normal weight, healthy diet, physical activity of at least 150 minutes per week and current non-smoking status [8]. This framework for preventing cardiovascular disease and promoting cardiovascular health is largely based on the principles of primary prevention [10]. The risk factors for CVD were categorized into two groups, with four ideal health behaviours, consisting of four behavioural risk factors including smoking status, normal weight, adequate physical activity, and healthy diet, and three biological factors, namely, blood pressure, cholesterol, and glucose. In 2022, the AHA updated its construct of cardiovascular health and included sleep as an additional behavioural risk factor; the construct which was conceptualized after completion of data collection for this study and has now been rebranded as "Life's Essential 8" instead of the former "Life's Simple 7" [11]. AHA stated that people can improve these factors through lifestyle and behaviour changes.

In urban Jamaica, ICH prevalence is low, with only 0.5% having all 7 characteristics and 23% having 5 or more in persons 20 years and older [12]. Other countries report a similarly low prevalence of ICH, ranging from 0.1% having all 7 ICH characteristics in the Republic of Serbia [13] to 3.3% having all seven ICH characteristics in Malawi [14].

Recent studies have suggested that having quality relationships is associated with better health outcomes, inclusive of cardiovascular disease and its risk factors [15–17]. Possible mechanisms for these associations include social support systems that may act as a buffer to stressors which compound an individual's medical condition [18, 19] Social support refers to the functions or provisions given by one's social relationships such as emotional concern, instrumental assistance, or information [18–20]. Social support involves an exchange of resources between a minimum of two individuals to benefit one of the individuals [21, 22]. It usually comprises the structural aspect of an individual's social life and the functions they seek to serve [23]. Social support becomes important in cardiovascular events as it acts as a buffer to the debilitating effects of stress induced cardiovascular reactivity [23]. Gottlieb emphasized that researchers in behavioural medicine and psychology have focused on social support for two main reasons [15]. Firstly, it plays an integral role in moderating the effects of life stress and well-being and secondly, it plays a key role in stress relief and adjustment in stressed individuals [15, 24]. Given these findings, social support ought to be a fundamental principle of

health promotion since it is expected to encourage healthier behaviours and potentially improve health outcomes [15].

A major concept in the application of social support is social embeddedness [25]. This refers to the degree to which individuals are interconnected and embedded within communities as described by Berkman and Krishna [26]. Social embeddedness is often measured by social network size, i.e., the number of persons with whom one is socially connected. Social network analysis usually focus more on the pattern of ties between persons in a social system rather than on the characteristics of the persons involved [26].

There is a growing interest among researchers regarding how social network factors interact with cardiovascular health [27]. Research has also shown that social networks can have an important effect on physical health and that among older adults, the number of friends (social network size) may influence cardiovascular health, with larger number of friends being associated with better CVD risks [18, 28]. Research has shown that lack of social support is associated with an increase in cardiovascular risk, with odds ratios ranging from 2 to 3 [29]. Higher social support has also been shown to be associated with reduced mortality, improvement in health and an overall greater quality of life [22]. In one study, Harding and colleagues found that high levels of functional social support was associated with lower risk of incident hypertension among participants in the Jackson Heart Study [30]. There is therefore evidence to support the notion that certain types of social support are associated with improved outcomes in cardiovascular conditions [30].

Although research has shown that social support may play an important role in the risk of cardiovascular diseases, little is known with regards to its relationship with cardiovascular health. Additionally, to our knowledge, no previous research on social support and cardiovascular health has been conducted in Jamaica. This study therefore aims to evaluate the relationship between social support and ideal cardiovascular health among persons living in urban Jamaica, a middle-income developing country.

## Methods

### Data sources

We conducted a primary cross-sectional study in urban communities from four parishes in Jamaica's southeast health region. The included parishes were Kingston, St Andrew, St Catherine, and St Thomas, representing almost 50% of the Jamaican population. S1 Fig shows a map of the included parishes and Jamaica's health regions. The included parishes provide a mix of urban communities, encompassing low-, middle-, and upper-income communities. The present study reports on one component of a study evaluating the impact of psychosocial stress, social networks, and social support on cardiovascular health in urban communities in Jamaica. Details concerning the research protocol and methods have been previously published [12, 31]. The data collection began on June 18, 2018, and ended on July 1, 2019. The study recruited participants from the 2016/2017 Jamaica Health and Lifestyle Survey (JHLS-III) or individuals living in one of the communities previously surveyed in JHLS-III and aged 15 years or older. We included non-institutionalized Jamaicans who were resident in the selected communities and excluded persons who were unable to comprehend the questions included in the questionnaire. We used a multi-stage sampling design with primary sampling units being randomly selected from Enumeration Districts (ED) in Jamaica. EDs are administrative divisions, consisting of approximately 100–150 households. Within each ED twenty households were selected using systematic sampling with sampling intervals based on the number of households in the ED and a random starting point provided by the Statistical Institute of Jamaica. Within each household one participant was selected using the Kish method [32]. This involved creating a schedule of all the occupants of the household starting with all males and then all females

from oldest to youngest. Based on the number of eligible individuals, a random person was selected from each household to complete the survey using a random selection table. We aimed to recruit 20 participants from each of 44 EDs for a total sample size of 880. We recruited 849 participants from 1130 persons approached resulting in a response rate of 75%.

The study was approved by the Mona Campus Research Ethics Committee of the University of the West Indies (Ethical approval # ECP 89 16/17). All participants, including those 15–19 years old, provided written informed consent, based on procedures approved by the Ethics Committee.

## Measurements

Data on demographic characteristics, cigarette smoking, physical activity and dietary practices were collected using interviewer administered questionnaires in face-to-face interviews. A copy of the questionnaire is available in the S1 Questionnaire.

Social support items for the questionnaire were adapted from the Chicago Community Adult Health Study (CCAHS). This questionnaire is now available in the public domain [33]. We followed the methods used by Viruell-Fuentes and colleagues [34] based on three questions:

i. network size based on answers to the question: "How many close friends and relatives do you have (people that you feel at ease with, can talk to about private matters, and can call on for help)?"

ii. instrumental support using the question: "How many friends and relatives do you have to whom you can turn when you need to borrow something like a household object or a small amount of money or need help with an errand?"

iii. informational support from the question: "How many friends and relatives do you have who you can ask for advice or information?".

All questions required a single numeric value. For descriptive analyses, the questions were reported separately. For analytic models, a score was created using Principal Component Analysis (PCA) as described below.

Blood pressure was measured using an oscillometric device (Omron 5 series blood pressure monitor, Omron Healthcare, Lake Forest, IL). These devices do not require recalibration but were checked regularly to ensure all components were functioning properly. Measurements were performed in the morning (typically between 7:00 am and 9:00 am). Three blood pressure measurements were taken using the right arm of the seated participant and followed standardized procedures developed for the International Collaborative Study of Hypertension in Blacks [35]. The means of the second and third systolic blood pressure (SBP) and diastolic blood pressure (DBP) measurements were used in the analyses. Weight was measured using a portable digital scale (Tanita HD-351 Digital Weight Scale, Tanita Corporation, Tokyo, Japan) and recorded to the nearest 0.1 kg. Height was measured using a portable stadiometer (Seca 213 Mobile stadiometer, Seca GmbH & Co., Hamburg, Germany) and recorded to the nearest 0.1 cm. To ensure reliable measurements, field staff were trained and certified to perform Blood Pressure (BP) and anthropometric measurements prior to starting fieldwork and at three-month intervals. Fasting glucose and total cholesterol were measured from a capillary blood sample using a point of care device (SD LipidoCare, Suwon, South Korea).

## ICH variables

The seven ICH categories were defined using the standards of the American Heart Association [7]. We did not have specific data on treatment for hypertension, diabetes mellitus or

hypercholesterolemia, so for these characteristics a report of doctor diagnosed was used instead of treatment for the condition. Normal blood pressure was defined as SBP less than 120 mm Hg and DBP less than 80 mm Hg among persons who were not previously diagnosed with hypertension. Normal body mass index (BMI) was defined as BMI less than 25.0 kg/m$^2$. Normal glucose was defined as a fasting glucose level of less than 5.6 mmol/L among persons who were not previously diagnosed with diabetes. Normal cholesterol was defined as less than 5.2 mmol/L among persons who were not previously diagnosed with hypercholesterolaemia. Current non-smokers were defined as persons who reported never smoking or have quit smoking for more than 12 months. Engaging in 150 minutes of moderate physical activity or 75 minutes of vigorous physical activity per week was defined as adequate physical activity for adults 20 years or older, while for participants 15–19 years adequate physical activity was defined as 60 or more minutes of physical activity per day. Participants were classified as having healthy diet if they had three or more of the dietary characteristics below:

- Low salt defined as no salt added at the table and rarely or never eating processed food.

- Low sugar-sweetened beverage (SSB) consumption defined as having SSB less than twice per week.

- Adequate fruit and vegetable consumption defined as having fruits or vegetables three or more times per day.

- Adequate fish consumption as having fish two or more times per week.

Except for physical activity, we used the same criteria to classify ICH characteristics for those 15–19 years and those 20 years and older. Although the AHA criteria suggest that BMI and BP for persons 12–19 years should be classified using the 85[th] and 90[th] centile respectively, we found that these cut points for those 15–19 years in our study would result in values higher than the cut points for normal used for adults (85[th] centile for BMI was 26.6 kg/m2, while the 90[th] centile for SBP was 127.9 mmHg and for DBP 81.8 mmHg).

Baseline socioeconomic status (SES) characteristics included participants' educational attainment; this was categorized as:

- "Less than high school" for persons with no formal education or up to grade 8

- "High school" for persons who would have attained formal education up to grades 9–13.

- "More than high school" for persons with post-secondary education at college, university, or vocational training institutions.

This was used as an individual level socioeconomic status [12].

Unimproved land value for each property for the communities selected for the survey were obtained from Jamaica's National Land Agency (NLA). Community property values were classified using the median property value for each community as a summary measure of the socioeconomic status for that community. The median property values were ordered and distinguished into rural and urban categories and then divided into tertiles, namely, lower, middle, and upper categories. Only urban communities were included in this study. Details on the use of this metric in the Jamaican context have been previously published [12].

Poverty was defined using the Planning Institute of Jamaica (PIOJ) Poverty Map data base [36]. Levels of poverty were estimated based on household consumption patterns quoted in Jamaican dollars per adult across 14 parishes and 767 communities of the country. This approach provided a Poverty headcount, estimated using a measure of consumption which compared the values of the food poverty line with the value of the total poverty line [36]. The

food poverty line represents the cost of the food basket which consists of a minimum nutritional requirement of kilocalories per family. The total poverty line consists of the food basket which is added to non-food items such as transport cost, clothing, health, educational and personal expenses [36]. For this analysis communities were ranked according to total poverty and categorized into lower, middle, and upper tertiles, such that those in the upper tertile have the highest level of poverty.

## Sample size and power

Sample size and power calculations were performed using Stata 16.1. We estimated the power based on an available sample of 841 study participants. To account for the complex survey design and the fact that the analyses will use survey analytic methods, we estimated a design effect factor and used this to estimate an effective sample size for the study [37, 38]. The design effect was estimated as the ratio of the variance for the proportion of participants with five ICH components in a previous national survey Jamaica Health and Lifestyle Survey 2007–2008 (JHLS-II) with and without adjustment for survey sampling design, as detailed in our study protocol [31, 39]. This was found to be 2.6; a similar estimate of the design effect was found when other outcomes were used. Based on this design effect, our effective sample size was estimated to 324 (i.e., 841/2.6) for this study. We did not have estimates of the variation of ICH by social support groups, but we estimated power based on possible odds ratios of 1.5 to 2.5 per unit increase in social support score (which is assumed to follow the Normal distribution while prevalence of ICH at mean support score is assumed to be 0.23). The effective sample of 324 has power of 81% to estimate an odds ratio (OR) of 1.5 using logistic regression ($\alpha$ = 0.05 two-sided) and 99% to detect an OR of 2.5 in similar models.

## Statistical approaches

Statistical analyses were done using Stata version 16.1. Community-specific data inclusive of land value and poverty data were identified from the National Land Agency (NLA) and the Statistical Institute of Jamaica (STATIN) respectively, these were merged into the primary study dataset. Given that social support was assessed using three correlated variables (network size, instrumental support, informational support), we used principal components analysis (PCA) to derive a score combining the three variables. Kaiser-Meyer-Olkin (KMO) measure of sampling adequacy was used to determine the suitability of the variables for use in PCA, with KMO greater than 0.5 seen as significant [40]. KMO for the three variables included was 0.739, thus adequate for use in PCA. Horn's parallel analysis method was then used to select the PCA components for use in the analysis [41, 42]. This yielded a single PCA component with Eigenvalue of 2.37 explaining 81% of the variance in the three variables. PCA scores were used in bivariate and multivariable analysis to assess the association between ideal cardiovascular health and social support.

## Outcome and exposure variables

The prevalence of all seven ICH characteristics have been found to be low in many populations, as such, investigators have used an operationalized definition of having five or more characteristics to determine ICH outcome as outlined in our study protocol and a previous publication from this study [12, 31]. Considering this we decided *a priori* to use having five or more ICH components as the main outcome variable in evaluating associations. The main exposure variable was social support score as defined above. Covariates and potential confounders assessed included age, sex, and measures of SES variables (education level, median community property value and community poverty).

T-tests were used to identify differences in means and Chi-squared and Fisher's exact used to identify differences in proportions. A p-value less than 0.05 was considered statistically significant for analyses. Logistic regression models were used to assess bivariate and multivariable associations.

**Survey weighted analyses.** Analyses were weighted to account for the multistage sampling used in the survey design. Sampling weights were calculated using the inverse of the probability of selecting a dwelling from which a single respondent was selected and corrected for the probability of refusal to participate in the study. Weights were further calibrated using post-stratification weights to correct for the distribution of the sample as compared to the distribution of the target population as done in previous national surveys [43, 44].

## Missing data

Missing data were handled using multiple imputation. The total dataset included 849 participants. Analyses were limited to 841 participants who had data on age, sex, and survey weights. Missing data for other variables were filled in using multiple imputation by chained equations using Stata software's suite of multiple imputation commands [45]. For the variables included in the analyses for this paper 727 participants had full data and 114 participants had data for at least one variable missing. The total fraction of missing data was 13.5%, with the largest proportion of missing for a single variable being 6.4%. We therefore chose to do 20 imputations based on recommendations from Graham et al [46].

## Inclusivity in global research

Additional information regarding the ethical, cultural, and scientific considerations specific to inclusivity in global research is included in the S1 Checklist.

## Results

Unweighted summary statistics for participants characteristics by sex are presented in Table 1. The analysed sample included 841 participants with an overall mean age of 47.6 ± 18.4 years; this included 562 females and 279 males. Weighted estimates are shown in S1 Table. S2 Table illustrates how the sample weights correct for differences in the age sex distribution of the sample when compared to the Jamaica's urban population.

The mean height was 165.5 ± 9.7 cm, and the mean weight was 78.5 ± 23.0 kg; females weighed significantly more than males (79.7 kg vs. 76.0 kg; p = 0.029). The mean overall BMI was 28.7 ± 8.4 kg/m$^2$; females (30.4 kg/m$^2$) had significantly higher BMI than males (25.4 kg/m$^2$), p≤0.001. Mean systolic blood pressure (SBP) was 130.9 ± 22.5 mmHg. For the diastolic blood pressure (DBP), the overall mean was 84.4 ± 13.5 mmHg with no significant variation by sex (p = 0.787). Mean fasting blood glucose level was 5.9 ± 2.31 mmol/l (106.3 ± 35.1 mg/dl). Mean total cholesterol level was 4.4 ± 1.2 mmol/l (170.1 ± 46.0 mg/dl); females (4.5 mmol/l [/174.0 mg/dl]) had significantly higher total cholesterol levels than males (4.2 mmol/l [/163.4mg/dl]), p = 0.002. The mean ICH score was 3.22 ± 1.28, with females having significantly lower scores than males. Mean PCA derived social support score was 0.0 ± 1.56 and was significantly higher for males.

There were no significant sex differences in SES characteristics. For education categories, 51.9% of persons had high school level, while 25.8% had more than high school education and 22.4% had less than high school education. The mean median land value was 1.63 million JMD overall [females, 1.64M vs males, 1.62M] (p-value for male vs. female = 0.890). The highest percentage (54%) of persons were in the lower land value category, 27% were in the middle category and 17% in the upper category. The highest proportion of persons were in the upper

**Table 1. Unweighted mean values and proportions for participant characteristics by sex.**

| Characteristics | n | Male | Female | Total | P-values for sex difference |
|---|---|---|---|---|---|
| | (Male/Female) | n = 279 | n = 562 | n = 841 | |
| **Continuous Variables** | | Mean ± SD | Mean ± SD | Mean ± SD | |
| Age (years) | 841 (279/562) | 46.6 ±19.2 | 48.2 ± 18.1 | 47.6 ± 18.4 | 0.246 |
| Height (cm) | 811 (268/543) | 173.0 ± 8.3 | 161.7 ± 8.0 | 165.5 ± 9.7 | <0.001 |
| Weight (kg) | 817 (269/548) | 76.1 ± 21.7 | 79.7 ± 23.5 | 78.5 ±23.0 | 0.029 |
| BMI (kg/m2) | 808 (267/541) | 25.4 ±7.2 | 30.4 ± 8.5 | 28.7 ± 8.4 | <0.001 |
| SBP (mmHg) | 822 (270/552) | 132.9 ± 21.1 | 129.9 ± 23.1 | 130.9 ± 22.5 | 0.067 |
| DBP (mmHg) | 822 (270/552) | 84.2 ±14.1 | 84.47 ±13.3 | 84.4 ±13.5 | 0.787 |
| Fasting Glucose | 799 (261/538) | | | | 0.191 |
| SI Units (mmol/l) | | 5.8 ± 2.0 | 6.0 ± 2.5 | 5.9 ±2.3 | |
| US Units (mg/dl) | | (104.5 ± 35.1) | (108.3 ± 44.1) | (107.0 ± 41.6) | |
| Total Cholesterol | 798 (262/536) | | | | 0.002 |
| SI Units (mmol/l) | | 4.2 ± 1.2 | 4.5 ± 1.2 | 4.4 ± 1.2 | |
| US Unit (mg/dl) | | (162.8 ± 44.9) | (173.2 ± 46.0) | (169.8 ± 46.0) | |
| Total ICH score* | 755 (250/505) | 3.4 ± 1.2 | 3.1 ± 1.3 | 3.2 ± 1.3 | 0.002 |
| Social support score (PCA) | 835 (277/558) | 0.29 ± 1.76 | -0.14 ± 1.43 | 0.0 ± 1.56 | <0.001 |
| **Categorical Variables** | | n (%) | n (%) | n (%) | |
| Education | 831 (278/553) | | | | 0.962 |
| Less than High School | | 61 (21.9) | 125 (22.6) | 186 (22.4) | |
| High School | | 144 (51.8) | 287 (51.9) | 431 (51.9) | |
| More than High School | | 73 (26.7) | 141 (25.0) | 214 (25.8) | |
| Median Land Value | 837 (276/561) | | | | 0.598 |
| Lower Tertile | | 146 (52.9) | 312 (55.6) | 458 (54.7) | |
| Middle Tertile | | 83 (30.1) | 150 (26.7) | 233 (27.8) | |
| Upper Tertile | | 47 (17.0) | 99 (17.7) | 146 (17.4) | |
| Community Poverty | 837 (276/561) | | | | 0.252 |
| Lower Tertile | | 109 (39.5) | 190 (33.9) | 299 (35.7) | |
| Middle Tertile | | 48 (17.4) | 114 (20.3) | 162 (19.4) | |
| Lower Tertile | | 119 (43.1) | 257 (45.8) | 376 (44.9) | |

*ICH = ideal cardiovascular health. Score = number of individual characteristics that each participant has. The score excludes participant with missing values of any of the seven ICH variables.

PCA = Principal Component Analysis

category of total poverty (44.9%), the middle category had 19.4% and the lower category had 35.7%.

Fig 1 shows the median distribution of social support/network characteristics for study participants. The median friend number size was 4, males had a median value of 5 and females had a median of 4 friends. For instrumental support, the median number was 3 persons, males had a median of 4 while females had a median of 3 persons. Median value for informational support was 4 overall; males had a median score of 5 and females had 4 persons.

Prevalence of the individual ICH components and proportion of those with five or more components (ICH-5) is shown in Table 2. The highest prevalence was seen for current non-smokers at 81% and lowest prevalence for healthy diet at 19%. A higher proportion of males (45%) had normal BMI compared to females (28%); males (52%) also had a higher proportion with adequate physical activity level compared to females (30%).

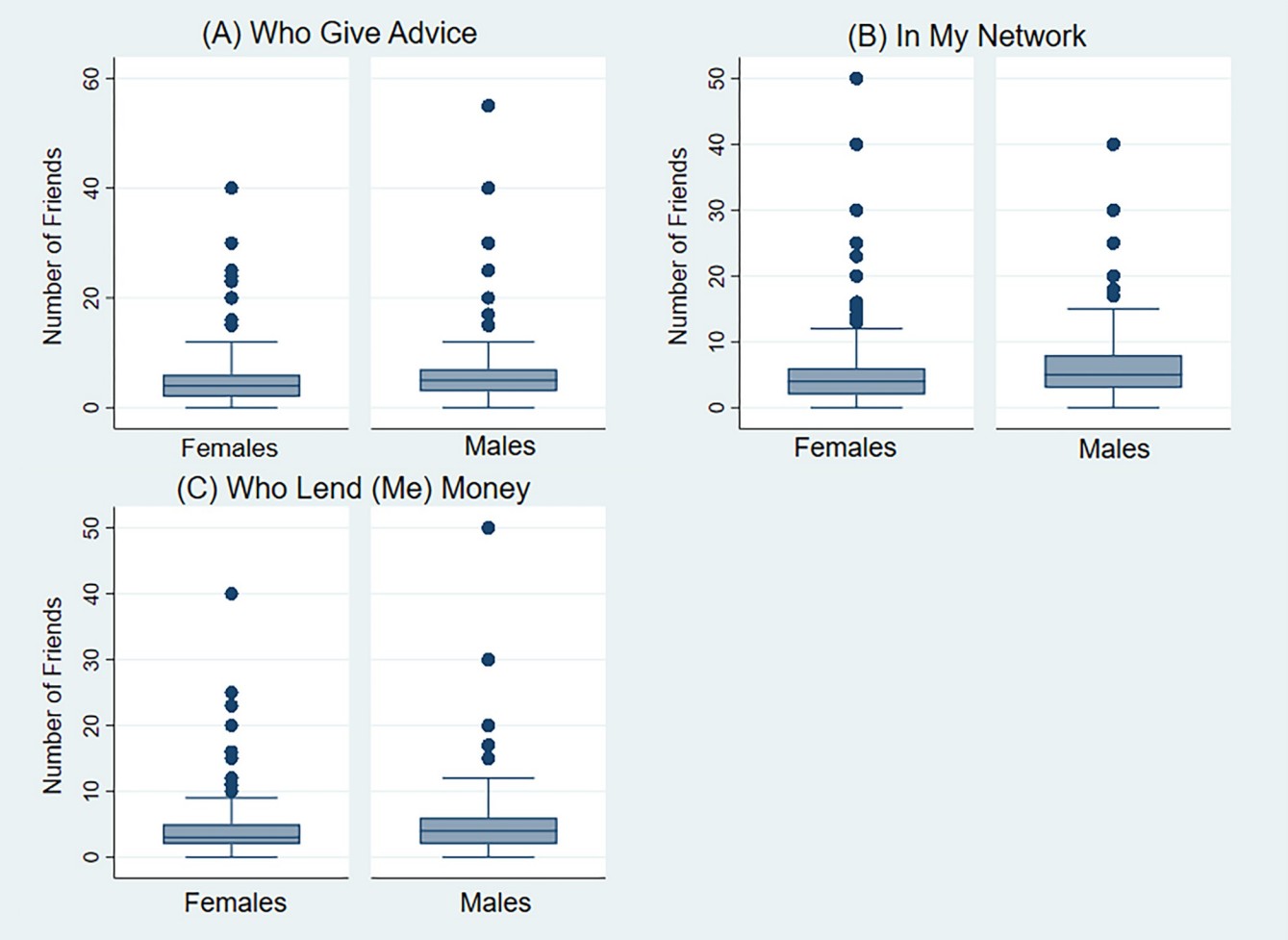

**Fig 1.** Boxplot with median and interquartile range for number of friends willing to give advice (A), number of friends overall (B), number of friends willing to provide a loan (C) for males and females.

The distribution of number of ICH variables and percent of participants with ≥1, ≥2, ≥3, ≥4, ≥5, ≥6 or 7 components is shown in Fig 2. The modal number of ICH variables was 3 with 31% of participants having 3 characteristics. None of the participants had all seven ICH characteristics. Prevalence of 5 or more components was 26.6% (95% CI 22.3–31.0); males 25.7%, females 27.5% (see Table 2). There was no significant difference in the prevalence of ICH by sex.

Prevalence of ICH-5 by tertiles of social support score and SES variables is shown in S3 Table. While there were no significant differences in the prevalence of ICH-5 by social support score tertiles for males and females combined, ICH-5 prevalence was highest among males in the lowest social support score tertile in sex-specific estimates. Prevalence of ICH-5 also showed significant associations with education and median land value for the total group and among males, but not among females. As shown in S4 Table, mean social support score varied significantly with SES characteristics.

Prevalence of individual ICH components by categories of social support score and SES characteristics is shown in S5 and S6 Tables. For education levels, significant associations were seen for non-smoker, normal glucose, normal blood pressure and normal cholesterol. For

**Table 2.  Prevalence estimates of ideal cardiovascular health (ICH) characteristics by sex.**

| Characteristics | Males + Females | Males | Females |
|---|---|---|---|
| | N = 841 | n = 279 | n = 562 |
| | % (95% CI) | % (95% CI) | % (95% CI) |
| BMI normal *** | 36 (32–40) | 45 (38–51) | 28 (23–32) |
| Non-smoker *** | 81 (77–85) | 72 (66–78) | 89 (86–92) |
| Glucose Normal | 68 (62–74) | 67 (59–74) | 68 (62–75) |
| Normal BP ** | 34 (29–40) | 29 (24–35) | 39 (33–46) |
| Adequate physical activity *** | 41 (36–45) | 52 (46–57) | 30 (24–37) |
| Healthy diet | 19 (16–22) | 16 (12–20) | 22 (18–26) |
| Cholesterol Normal | 79 (75–82) | 79 (73–85) | 78 (74–82) |
| ≥5 ICH Characteristics | 26.6 (22.3–31.0) | 25.7 (20.1–31.3) | 27.5 (22.8–32.3) |

CI = Confidence interval; ICH = ideal cardiovascular health

*p ≤0.05

**p ≤0.01

***p≤0.001

Normal BP = 120/80; Normal BMI < 25.0 kg/m$^2$; Normal glucose <5.6 mmol/L; Normal cholesterol <5.2 mmol/L; Adequate physical activity = 150 mins of moderate physical activity weekly or 75 mins of vigorous physical activity weekly; Non-smoking = never smoking or quit more than 12 months ago

median land value tertiles there were significant associations for normal BMI, adequate physical activity, and normal cholesterol. Normal glucose, healthy diet and normal cholesterol were associated with community poverty tertiles. When assessed by tertiles of social support score, the only statistically significant associations were for non-smoker and normal cholesterol among females (see S6 Table).

Table 3 displays odds ratios (95%CI) from bivariate analysis of the association between ICH-5, the outcome variable, and covariates among males and females combined and stratified by sex. We also report p-values for sex interaction. Odds of ICH-5 were lower among men with higher social support (OR (95% CI) = 0.69 (0.52–0.91); p = 0.010)) There was no significant association for females (p = 0.272). Men with high school education, had the higher odds of ICH-5 (OR (95% CI = 2.48 (1.48–4.15); p = 0.001). The odds of ICH-5 were again higher for men with more than high school education and for women with high school or more than high school education, but these were not statistically significant. Odds of ICH-5 were lower among men in the middle land value category ICH5 (OR (95% CI) = 0.46 (0.25–0.83); p = 0.012; associations in the other categories were not statistically significant. There was significant sex-interaction in the association between ICH and social support score (p = 0.003), therefore we used sex-specific multivariable models.

We also ran bivariate models for the individual ICH components as outcome variables with social support score as the exposure. The results for these models are shown in S7 Table. Higher social support score was associated with higher odds of being a non-smoker and having a healthy diet among females. Social support score was inversely associated with normal cholesterol among women. None of the associations achieved statistical significance among men.

Table 4 shows the final multivariable models of the relationship between ICH-5 and social support score. Again, we present sex-specific models as there was evidence for sex interaction in the relationship between ICH-5 and social support score. The models also included age, education category, land value and community poverty. Among females, social support score was directly associated with ICH-5, with odds ratio of 1.26 (95%CI 1.04–1.53). Age was significantly associated with ICH-5, odds ratio 0.94 (95% CI 0.91–0.97). Females in the upper category of land value had a significantly higher odds of ICH-5 (2.52; 95% CI 1.06–6.03).

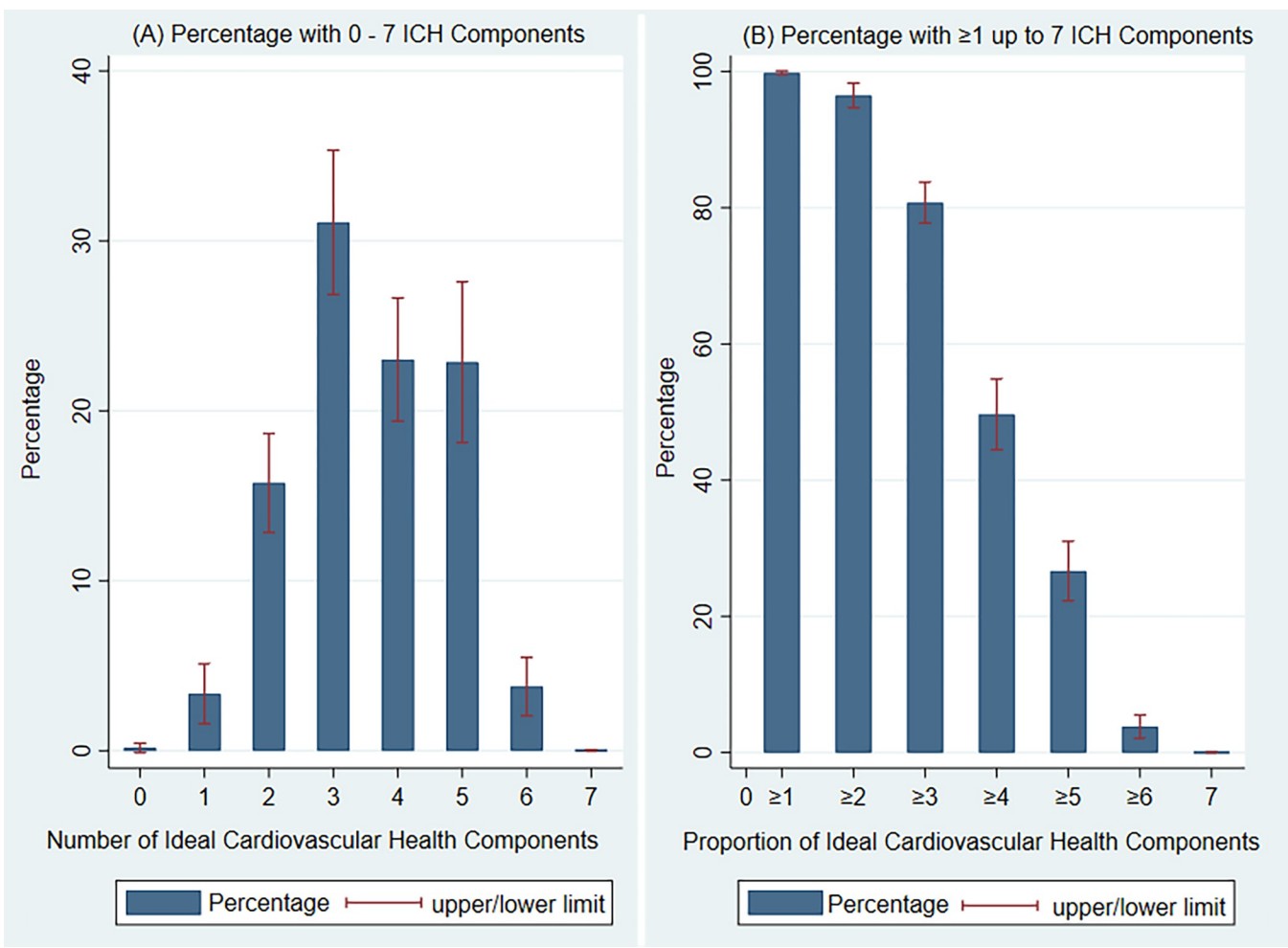

**Fig 2.** Proportion of participants with 0–7 Ideal Cardiovascular Health Components (A) and proportion of participants with ≥1, ≥2, ≥3, ≥4, ≥5, ≥6 or 7 Ideal Cardiovascular Health Components (B).

Among males, social support score was inversely associated with ICH-5, with odds ratio of 0.69 (95%CI 0.54–0.88). Age was significantly associated with ICH-5, odds ratio 0.96 (95% CI 0.93–0.99). Community poverty was directly associated with ICH-5 among men, middle category odds ratio 3.87 (95% CI 1.10–13.57); upper category odds ratio 2.04 (95% CI 1.04–3.98). There were no significant associations between education category and social support score among men or women.

In order to identify which of the ICH components were driving the observed associations, we reran the models using the individual ICH components as outcome variables, social support score as the exposure variable, and adjusting for age and SES variables as done in the final multivariable model specified above. These results are shown in S8 Table. Among women higher social support score was associated with higher odds of having normal BMI, being a non-smoker and having a healthy diet, but lower odds of having normal cholesterol. Among men higher social support score was associated with lower odds of being a non-smoker.

We also reran the final model using the dataset with no imputation to assess whether the use of imputed data would affect our findings or conclusions. These data are shown in S9 Table. Findings in the models without imputations were very similar to those reported with imputation and would result in similar conclusions.

**Table 3. Combined and sex-specific bivariate model showing the odds ratios for five or more ideal cardiovascular health characteristics (ICH-5) unit associated with change in age and social support score and categories of socioeconomic variables.**

| Variables | Males and Females | Males | Females | P-value for sex interaction |
|---|---|---|---|---|
| | Odds ratio (95% CI) | Odds ratio (95% CI) | Odds ratio (95% CI) | |
| Age | 0.96 (0.94–0.97) *** | 0.97 (0.94–0.99) ** | 0.95 (0.92–0.97) *** | 0.118 |
| Social Support Score | 0.87 (0.75–1.00) | 0.69 (0.52–0.91) * | 1.10 (0.98–1.29) | 0.003 |
| Education Categories | | | | |
| Less than High School | Reference | Reference | Reference | |
| High School | 2.15 (1.20–3.85) | 2.48 (1.48–4.15) ** | 1.88 (0.59–5.97) | 0.682 |
| More than High School | 1.89 (1.03–3.45 | 1.30 (0.61–2.76) | 2.50 (0.98–6.31) | 0.261 |
| Land Value Categories | | | | |
| Lower Category | Reference | Reference | Reference | |
| Middle Category | 0.54 (0.35–0.84) ** | 0.46 (0.25–0.83) * | 0.64 (0.39–1.05) | 0.311 |
| Upper Category | 0.95 (0.50–1.80) | 0.92 (0.42–2.03) | 0.98 (0.46–2.06) | 0.888 |
| Community Poverty | | | | |
| Lower Category | Reference | Reference | Reference | |
| Middle Category | 1.05 (0.67–1.64) | 1.18 (0.56–2.51) | 0.93 (0.48–1.80) | 0.656 |
| Upper Category | 1.82 (1.09–3.03) * | 2.00 (1.06–3.78) * | 1.65 (0.85–3.20) | 0.630 |

*p<0.05

**p<0.01; p<0.001

## Discussion

We have found that higher levels of social support, specifically larger number of friends, is inversely associated with ICH among males, but directly associated among women in urban Jamaica. Men had larger networks than women. Overall prevalence of ICH was low with none

**Table 4. Multivariable model[1] showing the odds of having five or more ideal cardiovascular health characteristics (ICH-5) for males and females with their corresponding 95% CI and p-value.**

| Variables | Females | P-value | Males | P-value |
|---|---|---|---|---|
| | Odds ratio (95% CI) | | Odds ratio (95% CI) | |
| Social Support Score | 1.26 (1.04–1.53) | 0.020 | 0.69 (0.54–0.88) | 0.005 |
| Age | 0.94 (0.91–0.97) | <0.001 | 0.96 (0.93–0.99) | 0.010 |
| Education Category | | | | |
| Less than High School | 1.0 | | 1.0 | |
| High School | 0.53 (0.11–2.61) | 0.422 | 1.20 (0.45–3.2) | 0.710 |
| More Than High School | 0.89 (0.25–3.24) | 0.861 | 0.64 (0.18–2.27) | 0.477 |
| Land Value Category | | | | |
| Lower Category | 1.0 | | 1.0 | |
| Middle category | 1.48 (0.68–3.21) | 0.305 | 0.46 (0.16–1.33) | 0.148 |
| Upper Category | 2.52 (1.06–6.03) | 0.038 | 2.49 (0.76–8.22) | 0.129 |
| Community Poverty Category | | | | |
| Lower Category | 1.0 | | 1.0 | |
| Middle Category | 1.07 (0.45–2.53) | 0.873 | 3.87 (1.10–13.57) | 0.035 |
| Upper Category | 2.46 (0.93–6.50) | 0.069 | 2.04 (1.04–3.98) | 0.038 |

[1]Separate models created for males and females with ICH-5 as outcome, social support score and main exposure variable and adjusting for age, education, median land value and community poverty. Models were weighted for survey design and used multiple imputation.

of the participants having optimal levels of all seven characteristics and only approximately a quarter of participants having five or more components. We also found that men in poorer communities were more likely to have five or more ICH components while women in communities with higher median property value were more likely to have five or more ICH components. These findings suggest that ICH was directly associated with social support and socioeconomic status among urban women in Jamaica, while on the contrary, ICH was inversely associated with social support and SES among Jamaican men.

Our findings with regards to the prevalence of ICH is consistent with previous studies. We previously reported a 23% prevalence for ICH among adults 20 years and older in a national urban sample [12]. In a Swiss study, the prevalence of three or more ICH characteristics was found to be 54%, while in a study in Peru only 12.7% had 5 or more ICH characteristics and no one had all 7 ICH characteristics [47, 48]. Similar to our findings, the study from Peru found that persons in the lower SES regions had higher percentage of ICH prevalence [48]. In other studies prevalence of five or more ICH characteristics ranged from 3.9% among men in Mississippi to 36.9% in Quebec [12]. With regards to social network size mean number of friends reported in our study was lower than that reported among US Latinos, where Viruell-Fuentes [34] reported mean sizes of 6, 5 and 5 for social network size, instrumental support and information support compared to median values of 4, 3 and 4 in our study.

One factor contributing to the low prevalence of cardiovascular health factors in Jamaica and other countries is the obesogenic environment, characterised by limited access to healthy foods and limited space for physical activity [49–54]. In Jamaica, Cunningham-Myrie and colleagues found that greater street connection density was associated with increased BMI among men and increased waist circumference was associated with increased distance from public parks among persons in the middle SES groups [49]. Additionally, further distance away from supermarkets was associated with higher mean BMI among women [52]. Guo and colleagues found that persons living in more obesogenic environments had greater odds of cardiovascular disease and were more likely to engage in unhealthy behaviours [54]. These studies suggest that efforts to improve cardiovascular health in Jamaica and similar populations should include measures to make the environment more conducive to healthy lifestyles.

We did not find many studies looking at social support and cardiovascular health, but in one study among US Latinos, Murillo and colleagues [55] found that larger central family size was associated with lower odds of healthy BMI and lower odds of having five healthy lifestyle factors (favourable BMI, physical activity, diet, alcohol use, smoking), while having a larger number of extended family members they felt close to was associated with higher odds of healthy BMI and being a non-smoker. While the findings among men in our study may seem counterintuitive, it is consistent with previous studies from our group showing higher prevalence of some cardiovascular disease risk factors among men [56, 57]. In these studies, obesity and metabolic syndrome were more prevalent among higher SES men. In contrast with the association with obesity and the metabolic syndrome, cigarette smoking was more prevalent among lower SES men [58]. While there has been an increasing body of work on social networks and cardiovascular disease, Child and Albert reported that there has been very little work in Black populations [59]. They further suggest that there may be important differences in the structure and function of social networks in Black populations. The findings from our study supports this notion and suggest that further research is necessary to advance our understanding in this field.

Gender differences in associations between social factors and health outcomes is a consistent finding in studies from Jamaica [12, 56, 60–62]. The precise aetiology is still unsure but may be related to differences in the determinants of behavioural risk factors such as smoking, diet, and physical activity. Physical activity tends to be higher in lower SES males who are

more likely to be employed in manual jobs and to use more active forms of transportation than higher SES males. Lower SES males are also less likely to consume highly processed foods and high fat foods. This leads to lower levels of obesity among lower SES males. On the other hand, higher SES women may be more likely to engage in leisure time physical activity and may be able to afford more healthy diets. Tobacco smoking is usually higher among lower SES men.

Gender differences in the impact of social support on ICH seems to be driven largely by differences in smoking behaviour and healthy diet. Higher social support was associated with higher odds of non-smoking and healthy diets among females, whereas the opposite seems to be true among men. It is likely that men with large social networks engage in more unhealthy behaviours whereas for women large social networks leads to more healthy behaviours.

With regards to the mechanisms linking social support and cardiovascular health, a number of studies have reviewed this relationship and have suggested pathways between social support and disease morbidity and mortality [17, 23, 63, 64]. Social support may influence health outcomes and contribute to better mental health and may also act as a buffer to stressful chronic conditions [23, 64]. A study by Bostean and colleagues highlighted that protective effects of social support from family and friends coupled with neighbourhood characteristics influences mental health by acting as a buffer to stressful life events [65]. Research has also revealed that social support possesses the ability to encourage positive reframing, thus reducing the stress [66]. Such stress may lead to high blood pressure which increases the risk of heart attack and stroke [67]. Additionally, Uchino and colleagues have found that higher levels of supportive family members was associated with lower hs-CRP, while higher numbers of ambivalent family members was associated with higher hs-CRP suggesting that social support may impact health through effects on inflammatory pathways [28]. Work by Kim and Thomas also reports lower levels of inflammation in persons with high social support thus supporting a role for inflammatory pathways in the mechanisms underlying social support and health [64]. Uchino also reported that social support has been linked to various cardiovascular, neuroendocrine, and immune system mechanisms which may lead to beneficial effects of social support on health. The perceived availability of social support may contribute to a positive sense of self, thus encouraging the accomplishment of personal task and influence behaviours [64]. These support systems may play an integral role in guiding individuals towards a level of optimum social performance since social support may manifest as ensuring one eats properly, exercise, avoid smoking and engage in other beneficial health behaviours.

We believe that this study contributes to the growing body of literature on the role of social support in health and helps to fill the research gap related to social support in Black populations. This study should also serve as a catalyst for more research in this field. The study also contributes to our understanding of the social determinants of health in a developing country context and highlights the need to assess gender differences and the need to consider sex-specific approaches when developing interventions to improve cardiovascular health.

Strengths of this study includes the following. The study sample is representative of urban communities in the Southeast region of Jamaica, which is the most populous areas of the country. The study used survey weights to ensure that estimates were representative of the region study. To our knowledge, this is the first study to evaluate the association between social support and cardiovascular health in the Jamaican context. We acknowledge however that some data were self-reported and as such may be subject to recall or reporting bias; however, we used standard questionnaire administration procedure to minimise these biases. We also note that the sample included more women compared to men as is commonly seen in local health surveys. Our use of survey weights would correct for this anomaly. We also note that we had missing data for some variables, but we were able to use multiple imputation to handle missing

data, thus minimizing potential loss of power and minimising bias. We were also able to show that the findings limited to participants with no missing data yielded similar results, so we are confident the use of imputation did not influence our findings or conclusions. Additionally, we did not have data on the quality of social support, hence could not determine whether this influenced the observed associations. Another limitation is that we did not have data on sleep duration and therefore could not include this in our cardiovascular health metric. We also acknowledge that given the cross-sectional study design we are not able to definitively establish temporal relationships between the exposure outcomes so we cannot make causal inferences.

## Conclusion

The association between social support score and ICH characteristics varies by sex, with an inverse association among men and positive association among women. Women living in communities with high property value were more likely to have ICH, while men living in poorer communities were more likely to have ICH. These findings suggest that gender specific approaches may be needed when designing social interventions to improve cardiovascular health.

## Supporting information

**S1 Checklist. Checklist for inclusivity in global research.**
(DOCX)

**S1 Fig. Map of Jamaica showing parishes and health regions.** The study was conducted in communities selected from Jamaica's Southeast Health Region (yellow-coloured parishes on the map). Base layer map obtained from Humanitarian Data Exchange (web link: https://data.humdata.org/dataset/cod-ab-jam); source map is licensed under a Creative Commons Attribution 4.0 International license.
(TIF)

**S1 Table. Mean values for participant characteristics by sex (estimates weighted for survey design; no imputation).**
(DOCX)

**S2 Table. Population and sample age-sex numbers and proportions for urban Jamaica.**
(DOCX)

**S3 Table. Prevalence of ICH-5 by social support score tertiles and by socioeconomic status (SES) categories.**
(DOCX)

**S4 Table. Mean social support score (from PCA) by socioeconomic status (SES) categories.**
(DOCX)

**S5 Table. Prevalence of ideal cardiovascular health (ICH) characteristics by socioeconomic status (SES) level.**
(DOCX)

**S6 Table. Prevalence of ideal cardiovascular health characteristics by social support tertile.**
(DOCX)

**S7 Table. Odds ratio for unit change in social support score for each ideal cardiovascular health characteristics in bivariate models.**
(DOCX)

**S8 Table. Odds ratio for unit change in social support score for each ideal cardiovascular health characteristics in multivariable models.**
(DOCX)

**S9 Table. Survey weighted multivariable model without imputations showing the odds of having five or more ideal cardiovascular health characteristics (ICH-5) for males and females with their corresponding 95% CI and p-value.**
(DOCX)

**S1 Dictionary. Variable names and codes for dataset used in the analyses.**
(DOCX)

**S1 Dataset. Dataset for social support paper in Stata format.**
(DTA)

**S2 Dataset. Dataset for social support paper in Excel format.**
(XLSX)

**S1 Questionnaire. Questionnaire used for data collection.**
(PDF)

## Acknowledgments

The authors would like to thank the Harvard T H Chan School of Public Health Lown Scholars Program for general support. The authors also thank the National Land Agency (NLA) for providing the property value data and the Statistical Institute of Jamaica (STATIN) for providing the shapefile. The authors would also like to thank the Ministry of Health and Wellness and the National Health Fund Jamaica for providing access to the Jamaica Health and Lifestyle Survey 2016–2017 (JHLS-III). They also would like to thank the Geoinformation Units from both STATIN and NLA. Thanks to the field staff, study participants and the CAIHR/ERU administrative staff and the unit driver for their support.

## Author Contributions

**Conceptualization:** Marshall K. Tulloch-Reid, Ishtar Govia, Rainford J. Wilks, David R. Williams, Trevor S. Ferguson.

**Data curation:** Joette A. McKenzie, Shelly R. McFarlane, Damian K. Francis, Novie O. Younger-Coleman, Trevor S. Ferguson.

**Formal analysis:** Alphanso L. Blake, Novie O. Younger-Coleman, Trevor S. Ferguson.

**Funding acquisition:** David R. Williams, Trevor S. Ferguson.

**Investigation:** Nadia R. Bennett, Joette A. McKenzie, Marshall K. Tulloch-Reid, Ishtar Govia, Shelly R. McFarlane, Renee Walters, Damian K. Francis, Rainford J. Wilks, Novie O. Younger-Coleman, Trevor S. Ferguson.

**Methodology:** Alphanso L. Blake, Nadia R. Bennett, Marshall K. Tulloch-Reid, Ishtar Govia, Shelly R. McFarlane, Rainford J. Wilks, David R. Williams, Novie O. Younger-Coleman, Trevor S. Ferguson.

**Project administration:** Joette A. McKenzie, Renee Walters.

**Supervision:** Novie O. Younger-Coleman, Trevor S. Ferguson.

**Writing – original draft:** Alphanso L. Blake, Trevor S. Ferguson.

**Writing – review & editing:** Alphanso L. Blake, Nadia R. Bennett, Joette A. McKenzie, Marshall K. Tulloch-Reid, Ishtar Govia, Shelly R. McFarlane, Renee Walters, Damian K. Francis, Rainford J. Wilks, David R. Williams, Novie O. Younger-Coleman, Trevor S. Ferguson.

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
