## [Decision Letter · Decision Letter 0]

3 Jan 2024

PGPH-D-23-02025

Social support and ideal cardiovascular health in urban Jamaica: a cross-sectional study

Dear Dr. Ferguson,

Thank you for submitting your manuscript to PLOS Global Public Health. After careful consideration, we feel that it has merit but does not fully meet PLOS Global Public Health’s publication criteria as it currently stands. Therefore, we invite you to submit a revised version of the manuscript that addresses the points raised during the review process.

We look forward to receiving your revised manuscript.

Kind regards,

Panniyammakal Jeemon

Academic Editor

Journal Requirements:

2.Please provide separate figure files in .tif or .eps format.

Additional Editor Comments (if provided):

Reviewers' comments:

Reviewer's Responses to Questions

**Comments to the Author**

1. Does this manuscript meet PLOS Global Public Health’s publication criteria? Is the manuscript technically sound, and do the data support the conclusions? The manuscript must describe methodologically and ethically rigorous research with conclusions that are appropriately drawn based on the data presented.

Reviewer #1: Partly

Reviewer #2: Partly

Reviewer #3: Partly

2. Has the statistical analysis been performed appropriately and rigorously?

Reviewer #1: Yes

Reviewer #2: Yes

Reviewer #3: No

3. Have the authors made all data underlying the findings in their manuscript fully available (please refer to the Data Availability Statement at the start of the manuscript PDF file)?

Reviewer #1: Yes

Reviewer #2: Yes

Reviewer #3: Yes

4. Is the manuscript presented in an intelligible fashion and written in standard English?

Reviewer #1: Yes

Reviewer #2: No

Reviewer #3: No

5. Review Comments to the Author

Reviewer #1: Thank you for an interesting paper on an important topic. I am happy to see the analysis that was undertaken and I think the findings can be relevant to adding to the literature as well as in policy recommendations. I am including some comments for your consideration.

Line 59, maybe more appropriate to use the term “social determinants of health”

Line 67-68 the wording is a bit strange, do you mean maintaining treatment targets?

Line 96. This is a long sentence, maybe a colon or split int two?

Line 107 should be “interact”

Paragraph starting at line 101 – this section is not very coherent. There are a lot of concepts and arguments at once. From here the introduction could be strengthened. I think a definition of concepts is important as a number of different terms are being used. It is also risky to refer to social networks when this could be confused with online platforms. Maybe fine a term you want to use throughout and stick with it?

Line 114 – there is some repetition here. I think a closer look at the wording here is important.

Line 122 – 123. You need to make a better distinction between cardiovascular diseases and what you refer to cardiovascular health. Do you mean cardiometabolic risk factors?

Line 124 – A statement like this needs references.

Methods

It would be helpful for readers unfamiliar with Jamaica to know a bit more about this region and these parishes. Are the urban? Rural? Mixed? What kind of populations?

Presumably the data are only self-reported for the first variables you mention? I would suggest separating two sections: self reported data; objectively measured data. Were any adjustments made to the CCAHS for the Jamaican population? Was this then turned into a scale? How much value is assigned to each question? Perhaps the questions and their score could be moved to a table for quick reference.

Line 154 – Did the blood pressure measurements all take place on the same day/time? Were point-of-care devices calibrated?

Line 168 – How did you ascertain treatment for hypertension or diabetes?

Line 238 – It would be helpful to know what proportion had the five or more components.

Line 273 – typo for repetition. The values described here could be maybe added to the table instead of written out?

Line 290 – No need to described what is in the table so directly. Maybe just note the main patterns instead.

I find Table 2 difficult to interpret. I think it would make more sense to show the ICH score by different categories of the values? So BMI categories (<25, 25-30, 30+), Smoking (current, non), Glucose (normal, elevated, diabetes) etc. It would help to get a sense if there is a dose type relationship for some of these variables.

I think there may also be too much text that directly describes the table for Table 3 starting from line 306. Maybe summarize the overall findings and trends rather than restating the table.

Discussion (line numbers have disappeared)

You say there is an association with age and ICH-5 but it is an inverse association. This is hardly surprising since people develop more NCDs as they age and perhaps are more likely not to have control than in earlier stages of disease. You need to say something more about these implications of these findings rather than just restating the results.

I think you are ignoring the role that an obesogenic environment plays including access to safe spaces for engaging in physical activity as well as to healthy foods. This has been extensively documented in Jamaica and I would expect to see this as part of the discussion. Alcohol use is also not included here which is high in Jamaica and especially among men. I would think this could also confound some of your findings.

There is also something that is not being mentioned which is the impact of mental health on risk behaviors like smoking and eating. This could also factor into sex differences as well as a differential affect across poverty groups. This should also be included.

Implications

You mention neighbourhood characteristics. I did not see this in the paper. There is the overall land value of the area but that is not a built environment or social characteristic and I think this statement is overreaching. This study provides important insight into the role of social support across different SES levels in Jamaica and certainly adds to the literature there. I think you can stick with that.

The strengths and limitations should be expanded. There are limitations to self-reported data and the imputation which are not being discussed.

Reviewer #2: Thank you for providing me with the opportunity to review the paper titled “Social support and ideal cardiovascular health in urban Jamaica: a cross-sectional study”. This paper by Alphanso et. al reports cross-sectional analysis of the association between social support (exposure variable) and ideal cardiovascular health (outcome variable defined as >=5 optimal levels of CVH factors). In general, the data reported, and multivariable logical regression analysis results are fine. However, the paper would benefit from more clarity around the methodological details, sampling strategy, and discussion of key findings, implications of study results, strengths and limitations.

General comments

• The basic underlying assumption in this study was that higher levels of social support leads to improved health behaviors, defined as ideal cardiovascular health (CVH). The authors examined the association between social support and ideal CVH among urban Jamaicans. I have several comments around the study methods.

• First, since none of the study participants achieved Ideal CVH as per the American Heart Association (AHA) guidelines based on 7 metrics, authors selected an arbitrary cut-off, i.e., >=5 levels as optimal CVH. It should be noted as a deviance from the standard definition of ideal CVH and justified in the methods section.

• Second, Social support was measured using a questionnaire that focused on social network size (number of friends/relatives), instrumental support (it is unclear whether information on the extent, or amount of support was also collected, which might indicate the strength of association by gender), and information support: were there any questions to assess the quality of information sources? Perhaps, the study tools can be provided in the appendix.

• Third, sample size: 841 participants: 279 males vs 562 females: I’m wondering why a higher proportion of women were sampled in this survey – what was the sampling frame/strategy, how did you ensure that the participants were randomly selected or to ensure a representative sample?

• Fourth, the authors found an inverse association for males and positive association between ICH5 and social support among females? What are the factors that explain these findings (sex related differences) needs to be interpreted in the local context – is it females are more influenced by social support in terms of adopting healthy behaviors due to network size, instrumental and informational support (can we say anything about the quality of information support). We need more explanation, possible pathways that link despite having higher access to information and instrumental support, males show lower levels of optimal health behaviors/ideal CVH.

• Fifth – how the study results might change in the light of recently introduced new CVH definition by the AHA – addition of 8th metric “Sleep” variable,

• CCAHS: Chicago Community Adult Health Study based social support questionnaire: was it validated for use in Jamaica?

Abstract:

• Participant’s age – mean, SD (add standard deviation)

Background

• Line: 74 – 78 – slight modification in the way AHA CVH is conceptualized: 4 behavioral risk factors including “smoking status” and 3 biological factors: blood pressure, glucose and cholesterol.

• Recently AHA added 8th dimension, Essential Life 8 (including sleep variable)

• A mention about the new essential life 8 metrics would be nice to make it more relevant and authors can explain the study was conducted prior to the introduction of CVH 8 metrics.

Methods and Design

• Survey design and sampling strategy requires more clarity

• How male and female participants from the same or different households were selected – I understand methods are described in some other paper, but would be useful to know, whether the study adopts simple random sampling or multistage cluster random sampling.

• What factors might explain skewed distribution of female and male participants included in this survey and analysis.

• Survey weights used? Unclear

• Sample size calculation – unclear what was the design effect factor used, and why do authors refer to sample size for each phase of the study – line 213.

Results

• Report standard deviation (SD) for all mean values. Refer line 261 – 269.

• FBG and cholesterol values could also be reported in mg/dl for international readers?

• Line 273 – delete repetition in “sex differences”

• Table 2. Remove commas after each value, either consistently provide all values with 1 decimal point or remove the 2 decimals provided for the total ICH score. Add definition or threshold as footnote in all Tables

• Descriptive analysis – bivariate analysis of social support score by major socio-demographic factors?

• Table 3 – why not show estimates for the overall population, in addition to presenting results stratified by men and women?

• This paper has too many Tables and Figures – I suggest moving some to the supplement. Or Figures 1-3 could all be plotted in a single panel.

Discussion

• What explains the a

---

## [Decision Letter · Decision Letter 1]

20 Jun 2024

Social support and ideal cardiovascular health in urban Jamaica: a cross-sectional study

PGPH-D-23-02025R1

Dear Dr. Ferguson,

We are pleased to inform you that your manuscript 'Social support and ideal cardiovascular health in urban Jamaica: a cross-sectional study' has been provisionally accepted for publication in PLOS Global Public Health.

Best regards,

Zulkarnain Jaafar

Academic Editor

Reviewer Comments (if any, and for reference):

Reviewer's Responses to Questions

**Comments to the Author**

1. If the authors have adequately addressed your comments raised in a previous round of review and you feel that this manuscript is now acceptable for publication, you may indicate that here to bypass the “Comments to the Author” section, enter your conflict of interest statement in the “Confidential to Editor” section, and submit your "Accept" recommendation.

Reviewer #1: All comments have been addressed

Reviewer #3: All comments have been addressed

2. Does this manuscript meet PLOS Global Public Health’s publication criteria? Is the manuscript technically sound, and do the data support the conclusions? The manuscript must describe methodologically and ethically rigorous research with conclusions that are appropriately drawn based on the data presented.

Reviewer #1: Yes

Reviewer #3: Yes

3. Has the statistical analysis been performed appropriately and rigorously?

Reviewer #1: Yes

Reviewer #3: Yes

4. Have the authors made all data underlying the findings in their manuscript fully available (please refer to the Data Availability Statement at the start of the manuscript PDF file)?

Reviewer #1: Yes

Reviewer #3: Yes

5. Is the manuscript presented in an intelligible fashion and written in standard English?

Reviewer #1: Yes

Reviewer #3: Yes

6. Review Comments to the Author

Reviewer #1: Thank you for addressing the comments. The paper is improved and in my view ready for the next step.

Reviewer #3: Thank you revising the manuscript based on comments.

7. PLOS authors have the option to publish the peer review history of their article (what does this mean?). If published, this will include your full peer review and any attached files.

**Do you want your identity to be public for this peer review?** For information about this choice, including consent withdrawal, please see our Privacy Policy.

Reviewer #1: No

Reviewer #3: **Yes: **Buna Bhandari
